# A squarate-pillared titanium oxide quantum sieve towards practical hydrogen isotope separation

Qingqing Yan[1,10], Jing Wang [2,10], Linda Zhang [3,4,9,10] ✉, Jiaqi Liu[5], Mohammad Wahiduzzaman [6], Nana Yan[2,7], Liang Yu[5], Romain Dupuis [6,8], Hao Wang[5], Guillaume Maurin [6], Michael Hirscher [3,4], Peng Guo [2,7] ✉, Sujing Wang [1] ✉ & Jiangfeng Du [1]

Separating deuterium from hydrogen isotope mixtures is of vital importance to develop nuclear energy industry, as well as other isotope-related advanced technologies. As one of the most promising alternatives to conventional techniques for deuterium purification, kinetic quantum sieving using porous materials has shown a great potential to address this challenging objective. From the knowledge gained in this field; it becomes clear that a quantum sieve encompassing a wide range of practical features in addition to its separation performance is highly demanded to approach the industrial level. Here, the rational design of an ultra-microporous squarate pillared titanium oxide hybrid framework has been achieved, of which we report the comprehensive assessment towards practical deuterium separation. The material not only displays a good performance combining high selectivity and volumetric uptake, reversible adsorption-desorption cycles, and facile regeneration in adsorptive sieving of deuterium, but also features a cost-effective green scalable synthesis using chemical feedstock, and a good stability (thermal, chemical, mechanical and radiolytic) under various working conditions. Our findings provide an overall assessment of the material for hydrogen isotope purification and the results represent a step forward towards next generation practical materials for quantum sieving of important gas isotopes.

Deuterium (D or [2]H), a stable isotope of hydrogen (protium or H), is well-known for its key importance in nuclear energy industry[1–3]. Lately, the worldwide extreme weather and the unexpected regional instabilities have displayed a far-reaching impact on the global energy system, pushing up natural gas and oil prices. This global energy crisis significantly promotes the growing demand of developing safe nuclear energy as one of the most feasible solutions[4–6], which requires a massive amount of deuterium with a high purity. Furthermore, unique

[1]Hefei National Research Center for Physical Sciences at the Microscale, Suzhou Institute for Advanced Research, CAS Key Laboratory of Microscale Magnetic Resonance, Hefei National Laboratory, University of Science and Technology of China, 230026 Hefei, China. [2]National Engineering Research Center of Lower-Carbon Catalysis Technology, Dalian National Laboratory for Clean Energy, Dalian Institute of Chemical Physics, Chinese Academy of Sciences, 116023 Dalian, China. [3]Max Planck Institute for Intelligent Systems, D-70569 Stuttgart, Germany. [4]Advanced Institute for Materials Research (WPI-AIMR), Tohoku University, Sendai 980-8577, Japan. [5]Hoffmann Institute of Advanced Materials, Shenzhen Polytechnic, 518055 Shenzhen, China. [6]ICGM, Univ. Montpellier, CNRS, ENSCM, Montpellier, France. [7]University of Chinese Academy of Science, Bejing 100049, China. [8]LMGC, Univ. Montpellier, CNRS, Montpellier, France. [9]Present address: Frontier Research Institute for Interdisciplinary Sciences (FRIS), Tohoku University, Sendai 980-0845, Japan. [10]These authors contributed equally: Qingqing Yan, Jing Wang, Linda Zhang. ✉e-mail: l.zhang@is.mpg.de; pguo@dicp.ac.cn; sjwang4@ustc.edu.cn

advantages of deuterium over protium have been discovered and intensively explored over the past decades in a wide range of applications including medical treatment, medicine revolution, optical fiber industry, national defense and scientific research[7–11], gaining rapidly increased attention. However, the natural abundance of deuterium is as low as 0.0156% in all hydrogen isotopes on the earth, implying the essential value of techniques to separate and enrich deuterium from the hydrogen isotopic mixture. Conventional industrial processes for deuterium purification such as electrolysis of heavy water, chemical exchange and cryogenic distillation have to face inevitable limitations such as complex, low efficiency, time- and energy-consuming operating conditions[12,13], which calls for alternative methods to approach the requirements of a sustainable deuterium production. Among them, kinetic quantum sieving (KQS) using porous materials to preferentially separate heavier deuterium over protium has aroused a great interest. It features either notably elevated separation efficiencies or desirable deuterium capacities, even approaching a good balance of efficiency and capacity in the case of some advanced porous materials, standing among the most promising alternative technologies for hydrogen isotope separation[12–15].

Crystalline and amorphous materials with either structurally ordered or randomly distributed porosities have been assessed their potentials in the KQS of hydrogen isotopes. It was demonstrated that KQS effect could be boosted using crystalline materials possessing chemically tunable cavities, for instance, two-dimensional (2D) crystals, porous organic cages (POCs), covalent organic frameworks (COFs) and metal-organic frameworks (MOFs). Furthermore, different strategies have been applied, such as using porous materials with open metal sites or flexible structures, to address the general trade-off issue between the $D_2/H_2$ selectivity and the $D_2$ adsorption capacity. All those valuable results suggested that a rigid ultra-microporous framework with a suitable local flexibility should be privileged, since an overall framework flexibility (breathing and swelling) is hard to control and makes the achievement of quantum effects elusive under operating conditions[16,17]. Previous research in this field mainly focused on the fundamental understanding of the mechanism in play and the establishment of the structure–performance relationship. It, therefore, not only laid a fundamental foundation to guide the rational structure design of the next generation material, but also indicated the required characters towards practical applications that a new material is supposed to feature, including a good balance between separation selectivity and working capacity, sustainable and scalable synthesis, long-term stability for easy handling and working condition, feasible regeneration and cycling.

Here we present an ultra-microporous inorganic–organic hybrid framework constructed from infinite titanium-oxide $(Ti_2O_3)_n$ 2D layer and rigid squarate linker through strong covalent bonding between $Ti^{4+}$ ion and hydroxyl group, denoted as USTC-700 (USTC, University of Science and Technology of China), that possesses good quantum sieving properties for deuterium separation and addresses the necessary material features outlined above. This compound displays zero thermal expansion in a wide temperature range (10–548 K), which leads to good $D_2/H_2$ separation performances combining a high selectivity and a good volumetric deuterium uptake with reversible adsorption-desorption isotherms. In addition, USTC-700 not only exhibits good thermal, chemical, mechanical, and radiolytic stabilities, but also can be produced by one-step reflux reaction under water-based green and scale-up conditions with industrial feedstock. The overall advantage combination makes USTC-700 among the most promising candidates towards the practical separation of hydrogen isotopes by KQS.

## Results and discussion
### Design and synthesis
Porous inorganics, such as oxides and molecular sieves, are recognized to have limited tunability of the pore characters (size, shape, and chemical environment), leading to difficulties in fine-tuning the corresponding KQS performance. On the contrary, organic-containing compounds have shown good structural tunability and tolerance, providing a new avenue to design material structure according to application request. To achieve a suitable balance of rigidity and flexibility, design and synthesis of structurally adaptable inorganic-organic hybrids would be the most straightforward strategy. However, the current strategy to prepare organic or hybrid rigid framework with ordered porosity mainly relies on the usage of organic molecules with six- or five-membered aromatic ring moieties. It leads to an unambiguous difficulty of forming ultra-micropores suitable for optimum KQS of deuterium[14], which calls for new types of inorganic building blocks and organic linkers to solve this problem. Accordingly, we anticipated that infinite $TiO_x$ building units (i.e., 1D chain and 2D layer) and squaric acid (semiconductor and laser industry feedstock) would be ideally assembled to generate the desired pore structures owing to the following reasons: i) $TiO_2$ is one of the most chemically stable inorganic compounds, showing long-term resistances under normal acidic, basic and hydrothermal conditions; ii) infinite $TiO_x$ building units, in particular the 2D layers, not only hold the similar structural rigidity and mechanical stability to that of the bulk $TiO_2$ along the direction of structure extension, but also assure the suitable distances between adjacent $Ti^{4+}$ ions to generate ultra-micropores when assembled with short linkers; iii) squaric acid is well-documented for its coplanar rigidity with local flexibility of its square moiety, as well as its tendency to construct ultra-microporous materials with different metal ions[18,19]; iv) the covalent bonding between squarate oxygen and metal ion is much stronger than those of the carboxylate-metal and nitrogen-metal ones, even in the cases of divalent metal ions (i.e., $Co^{2+}$, $Zn^{2+}$, $Mg^{2+}$, $Ca^{2+}$). As a result, elevated hydrothermal, chemical and thermal stabilities were observed for most of the metal-squarate coordination polymers[18–22].

It is worthy to note that synthesis of crystalline open framework based on $Ti^{4+}$ is one of the most challenging tasks in the field of coordination chemistry, due to the well-known difficulty of controlling the titanium reactions in solutions[23]. Except for Ti-phosphonate hybrids[24], the presence of excess water in other reaction systems preferentially facilitates the hydrolysis of water-sensitive titanium precursor to $TiO_2$, and impede the assembly of the $Ti^{4+}$ ions with organic linkers. Hence, it excludes the possibility to use water as the major reaction solvent. As a result, reported titanium open frameworks built with carboxylate and hydroxyl linkers were all prepared in harmful organic solvents and using air/water-sensitive Ti precursors, not only leading to costly synthesis and activation of material, but also hardly approaching the green chemistry level. Following the above guidelines, high-throughput screening of reaction conditions using various titanium precursors and squaric acid as reactants was conducted. In particular, searching green and scalable reaction condition is the major target. A highly crystalline product (USTC-700) was initially obtained in diluted hydrochloric acid (HCl, 3 M) with titanium oxyacetylacetonate $(TiO(acac)_2)$ precursor under hydrothermal conditions. Further optimization showed that various Ti precursors functioned similarly to generate the product (Supplementary Fig. 1) under reflux conditions with ambient pressures. It indicated the strong and fast interaction between the squarate with $Ti^{4+}$ ion, which succeeded in competition with the hydrolysis in diluted HCl. The squarate oxygens are so active towards $Ti^{4+}$ ion that it allows the use of inert industrial feedstock, titanium oxysulfate $(TiOSO_4)$, as the reactant. As expected, the synthesis of USTC-700 could be scaled-up using $TiOSO_4$ under reflux condition in diluted HCl (Supplementary Note 1). Therefore, it is a green and practical production of titanium open framework under mild conditions purely using cost-effective and stable feedstock as both reactants and solvent, which represents a clear leap forward in the discovery of Ti-based materials.

The crystallographic structure of USTC-700 was determined by one of advanced 3D electron diffraction (ED) technique called continuous

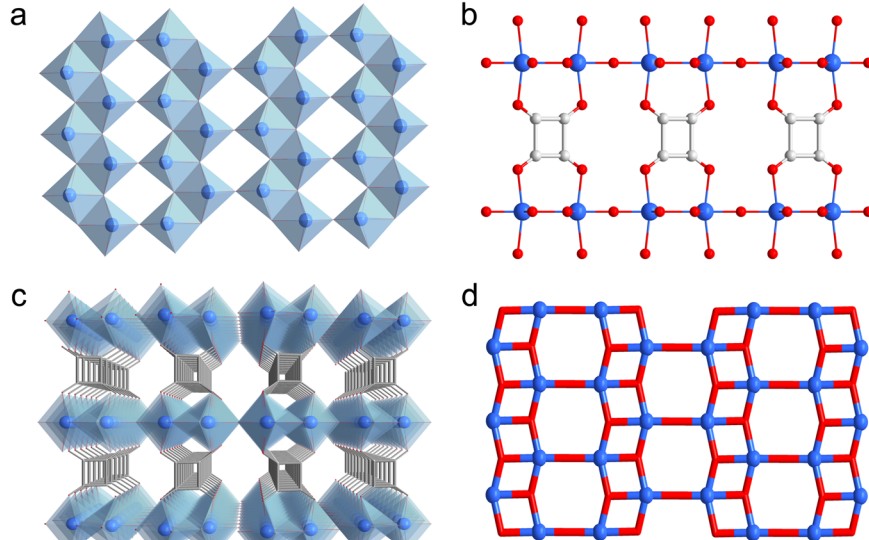

**Fig. 1 | Crystallographic structure of USTC-700 and the comparison with TiO$_2$(B). a** One infinite (Ti$_2$O$_3$)$_n$ 2D layer in USTC-700 structure. Double-strand zigzag titanium oxide chains show octahedral corner- and edge-sharing mode while adjacent chains adopt octahedral corner-sharing mode to connect each other, generating the 2D layers. The condensation degree of the layer is 1.5, among the largest ones in titanium-based coordination compounds. **b** Squarate linker molecules function as tetradentate pillars to bridge two pairs of Ti$^{4+}$ ions in the 2D layers above and below. **c** 3D framework of USTC-700 viewed along the *a*-axis, showing the ultra-micropores between squarate linkers. **d** A 2D layer in TiO$_2$(B) structure showing the same arrangement of titanium atoms and Ti-O connections as those in USTC-700.

rotation electron diffraction (cRED)[25] and further confirmed by powder X-ray diffraction (PXRD) (Supplementary Fig. 2, Supplementary Tables 1–3, and Supplementary Note 2). It was found that USTC-700 is of an unreported structure with a formula of [Ti$_2$(μ$_2$-O)(μ$_3$-O)$_2$(C$_4$O$_4$)] crystallizes in a monoclinic *P*2/m (10) space group with the unit-cell parameters of $a = 3.6974(2)$ Å, $b = 7.1931(5)$ Å, $c = 6.4272(5)$ Å, $\beta = 106.633(4)°$, and $V = 163.78(2)$ Å$^3$. Each Ti$^{4+}$ center is in an octahedral coordinative environment, in which one μ$_2$-oxo group and three μ$_3$-oxo groups form the equatorial plane while two squarate oxygens locate in the vertical poles. Neighboring Ti$^{4+}$ ions show a zigzag arrangement and connect each other by μ$_3$-oxo groups to share their octahedron corners and edges forming double-strand titanium oxide chains. Adjacent chains are further linked by μ$_2$-oxo groups to adopt an octahedron corner sharing mode to generate the infinite (Ti$_2$O$_3$)$_n$ 2D layers (Fig. 1a). Each squarate linker molecule stretches perpendicularly in the space between the adjacent layers, using its tetradentate oxygen atoms to bridge two pairs of Ti$^{4+}$ ions above and below separately (Fig. 1b). The 3D framework is generated by pillaring (Ti$_2$O$_3$)$_n$ layers with squarate spacers, featuring ultra-micropores of a square shape (3.1 Å × 3.1 Å in dimension) running along the *a*-axis (Fig. 1c and Supplementary Fig. 3), in line with the pore size distribution (2.6–3.2 Å) calculated from the density-functional theory optimized crystal structure (Supplementary Fig. 4 and Supplementary Note 4).

It is noteworthy that the same 2D layer of titanium oxide could be found in the crystal structure of TiO$_2$(B) compound (Fig. 1d), which crystallizes in monoclinic *C*2/m space group with unit-cell parameters of $a = 12.16$ Å, $b = 3.74$ Å, $c = 6.51$ Å and $\beta = 107.3°$[26]. The condensation degree (ratio of oxo to Ti ion) of the (Ti$_2$O$_3$)$_n$ layer is as high as 1.5, same as that of the reported record in MIP-177-HT[27] and very close to that in TiO$_2$. Therefore, the structure of USTC-700 maintains the characters of TiO$_2$(B) and could be seen as exfoliated TiO$_2$(B) layers pillared by squarates.

### Hydrogen isotope separation

The D$_2$/H$_2$ separation properties of USTC-700 were investigated by a comprehensive analysis of single component and mixed gas adsorption (Supplementary Note 3), including physisorption data, cryogenic

thermal desorption spectroscopy (TDS) and dynamic breakthrough measurement results, which are shown in Fig. 2.

Single component gas sorption of H$_2$ and D$_2$ on activated USTC-700 sample were carried out at different temperatures, ranging from 30 K to 77 K (Fig. 2a, b, Supplementary Fig. 5). The adsorption isotherms for both H$_2$ and D$_2$ show a typical type I behavior with the highest uptake for the lowest measurement temperature and decreasing values for higher temperatures. For all the exposure temperatures, the D$_2$ uptakes are higher than those of H$_2$, highlighting a higher affinity and more favorable diffusion of D$_2$ vs H$_2$ in the pores of USTC-700, since D$_2$ possesses a smaller de Broglie wavelength and lower zero-point energy in the confinement. At 1 bar and 77 K, the uptake of D$_2$ is 4.6% higher than that of H$_2$, while at 30 K the D$_2$ uptake is 11.8% larger, which indicates the dominant influence of quantum effect at lower temperature range. In addition, the uptake difference increases with decreasing equilibrium pressure and surface coverage. For example, at 1 mbar, the D$_2$ uptake is 15% higher than that of H$_2$ at 30 K, suggesting a thermodynamics quantum cryo-sieving effect. However, this effect is negligible compared with the effect of diffusion kinetics.

Notably, the width of the 1D pore channel of USTC-700 (2.6–3.2 Å) is comparable to the kinetic diameter of the H$_2$ molecule and the reported optimum pore diameter for quantum sieving (3.0–3.4 Å)[28]. However, the effective size (or the thermal de Broglie wave length) of the H$_2$ isotopes is inversely correlated to the temperature. Thus, the entire pore channel of USTC-700 can be regarded as a quantum sieve that would create a distinct penetration energy barrier for each hydrogen isotope—depending on the applied temperature. At the same time, the same pore surface might act as adsorption sites for the H$_2$ isotopes. Since USTC-700 framework lacks any particular binding sites on its internal pore surface, possible discrimination of the two isotopes via thermodynamics (interactions) can be discarded. Indeed, the calculated average $Q_{st}$ values applying the Clausius–Clapeyron equation were found to be 4 kJ mol$^{-1}$ and 3.5 kJ mol$^{-1}$ for D$_2$ and H$_2$, respectively (Supplementary Fig. 6).

D$_2$/H$_2$ selectivity was directly measured by the in-house built cryogenic TDS, using a 1:1 H$_2$/D$_2$ mixture (Supplementary Figs. 7-9 and

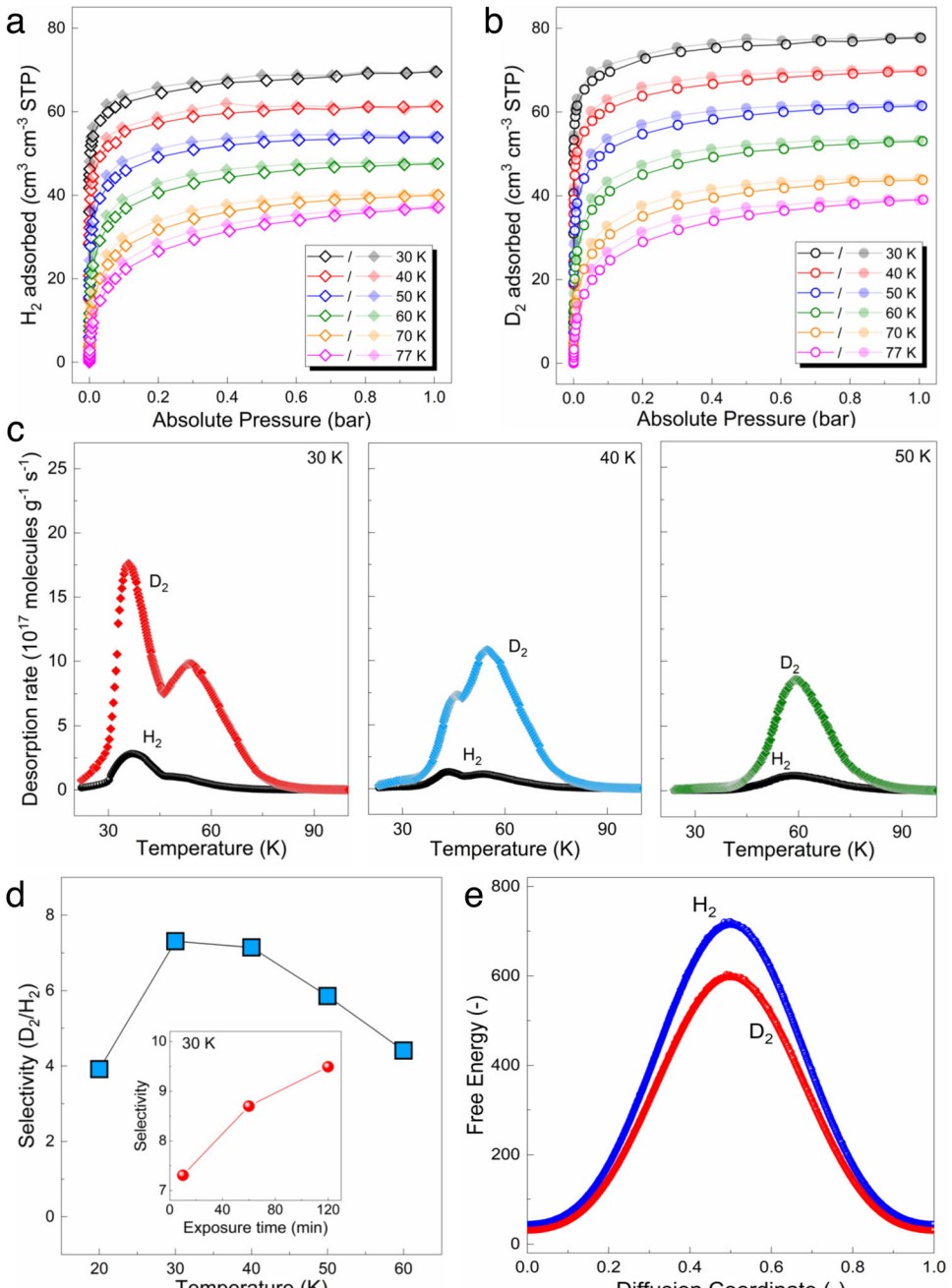

**Fig. 2 | Hydrogen isotope separation performance of USTC-700. a** $H_2$ single component adsorption (open symbols) and desorption (closed symbols) isotherms collected at different temperatures. **b** $D_2$ single component adsorption (open symbols) and desorption (closed symbols) isotherms collected at different temperatures. **c** Cryogenic thermal desorption spectroscopy (TDS) results obtained after exposure to a 10 mbar 1:1 mixture of $H_2$ and $D_2$ for 10 min at varied temperatures. **d** $D_2/H_2$ selectivity as a function of exposure temperature in TDS measurements. Inset, $D_2/H_2$ selectivity as a function of exposure duration at 30 K. **e** Free energy profiles of $H_2$ and $D_2$ for the plausible diffusion pathway along the 1D channel of the USTC-700 structure.

Supplementary Note 3). TDS spectra collected after a 10-min exposure to a 10 mbar mixture at different temperatures (30, 40, and 50 K) are presented in Fig. 2c. The $D_2/H_2$ selectivity summarized in Fig. 2d reveals a maximum of 7.3 at 30 K, while the corresponding $D_2$ uptake decreased from 1 mmol g$^{-1}$ at 30 K to 0.16 mmol g$^{-1}$ at 60 K. The inset shows the selectivity increases with longer exposure time at 30 K, from 7.3 to 9.5 after 120 min exposure. While, $D_2$ adsorption amounts increase slightly with increasing exposure time. This indicates that $H_2$ molecules are replaced gradually by the adsorbed $D_2$, and the $D_2/H_2$ selectivity increases markedly with longer time at the same temperature and pressure due to thermodynamics equilibrium. It is worthy to

note that working pressure did not show notable effect on the selectivity. A wide pressure range of 10 mbar to 300 mbar of the applied $H_2$/$D_2$ mixture all leads to constant selectivity (Supplementary Fig. 8), which makes USTC-700 a promising potential candidate towards practical conditions.

To further validate the capability of USTC-700 for $D_2/H_2$ separation, column breakthrough measurements were carried out with feeds of two different mixtures: $H_2/D_2/Ne$ (1/1/98, vol%) and $H_2/D_2/Ne$ (10/10/80, vol%). The experiments were performed on a column filled with ~1 g of the activated sample at 77 K and 1 bar, with a total gas flow of 5 mL min$^{-1}$. As shown in Supplementary Fig. 9, USTC-700 can effectively

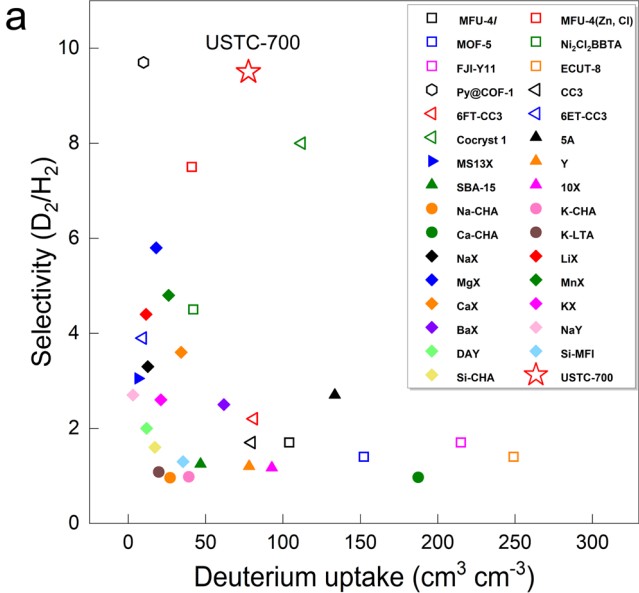

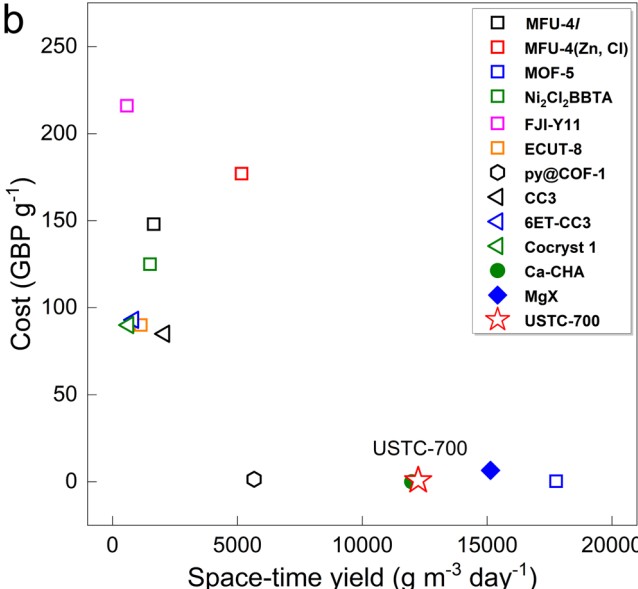

**Fig. 3 | Summary of separation performances and production parameters for various KQS materials. a** Comparison of volumetric deuterium uptake calculated using crystal density and KQS selectivity of USTC-700 and reported MOFs, COFs, POCs and zeolites. The best performance of each material was selected for comparison (see detailed condition in Supplementary Table 4). **b** Space-time yield and the cost to synthesize one gram of product for USTC-700 and typical synthetic KQS materials (Data were taken from literature, and were summarized in Supplementary Tables 4–6).

separate $H_2$ and $D_2$ for a mixture of $H_2/D_2/Ne$ (1/1/98, vol%). $H_2$ elutes out from the column at the 17th min $g^{-1}$ while $D_2$ is retained in the column for as long as 28 min $g^{-1}$. The notably longer retention time of $D_2$ than that of $H_2$ indicated the former was preferentially adsorbed by the sorbent, consistent with the single-component adsorption results. The breakthrough curve for a feed of $H_2/D_2/Ne$ (10/10/80, vol%) further confirmed that the separation ability of USTC-700 was fully maintained for higher concentrations mixtures (Supplementary Figs. 10 and 11). Furthermore, a good cycling performance was observed in column breakthrough measurements of USTC-700 (Supplementary Fig. 12).

As discussed earlier, the channel size of USTC-700 is expected to be the most critical parameters in determining the diffusion kinetics

and overall separation performances. To confirm this assumption, the free energy barriers for $H_2$ and $D_2$ molecules at 30 K (at which we observed the best experimental $D_2/H_2$ separation performance) to cross the channel entrance of USTC-700 were calculated using Monte Carlo simulations in the canonical ensemble and the application of the Widom's test particle insertion method (Supplementary Note 4)[29]. The so-called Feynman–Hibbs semi-classical effective potentials were used to account for the NQE in this force field based simulations[30]. These calculations revealed a significantly higher energy barrier for $H_2$ compared to $D_2$ (Fig. 2e), confirming that $H_2$ passing through the channel entrance of USTC-700 is more difficult than $D_2$ at this temperature.

Further, in order to probe the nuclear quantum effects on the confined $H_2/D_2$ molecules, path integral molecular dynamics calculations were performed for both $H_2$/USTC-700 and $D_2$/USTC-700 by accounting the quantum mechanical nature of both the electrons and the nuclei explicitly[31,32]. The gyration radius of the so-called path integral ring polymer[33] is a measure of the effective sizes of $H_2$ and $D_2$ molecules and how they are affected by the confinement of the material channel, which can be qualitatively related to the $D_2/H_2$ separation ability of USTC-700. Indeed, at 30 K, the gyration radius of the confined $D_2$ (0.47 Å) was found to be smaller than the confined $H_2$ (0.64 Å) within the USTC-700 channel, which is significant in regards to the solid pore size. Therefore, the pore confinement of the material has a significant destabilizing effect on $H_2$ compared to $D_2$, paving the way towards a preferential $D_2$ adsorption in USTC-700, in line with the experimental trend.

**Feasibility towards practical applications**

While reported studies on KQS materials mostly focused on their separation performances and the fundamental understanding of structure-performance correlations, key factors of materials themselves required for practical handling under working conditions, such as stabilities, cycling abilities, production costs, and space-time yields (STY), are yet to be considered and achieved, in particular for those synthetic compounds with no commercial availability. However, the evaluation of a material towards practical application should be adequately comprehensive to take account of as many parameters as possible, especially in terms of material's durability and scalability. Therefore, we carried out a broad analysis of all the aforementioned factors of USTC-700 in comparison with those of the reported KQS materials. The corresponding results are presented in Figs. 3 and 4.

Practical separation performance tends to address balancing $D_2/H_2$ selectivity and deuterium uptake to approach a good combination, which was well-illustrated by the encouraging results achieved with POCs published very recently[14]. Due to the size and shape requirements of apparatus, volumetric uptake of a sorbent is often more practically considered rather than the gravimetric one. As shown in Fig. 3a, USTC-700 displays one of the best $D_2/H_2$ selectivity among those porous materials with rigid structures, which is comparable to the record offered by the functionalized COF-1. On the other hand, USTC-700 shows a decent deuterium uptake of about 80 cm³ cm⁻³, which indicates a high adsorbing efficiency of the ultra-micropores in the framework structure. In particular, the reversible adsorption-desorption character of USTC-700 ensures facile working cycles and a good working capacity, which evidently outperforms most of materials involved for comparison (Supplementary Table 4). Therefore, USTC-700 is one of the best KQS materials to combine selectivity and volumetric uptake.

Scalability of the material production is an essential criterion to be considered for further stages beyond the proof of concept one in lab-scale, which is mainly reflected by production cost, synthesis method and space-time yield. To compare the overall cost to produce one gram of each synthetic sorbent, the lowest prices of major chemicals used for the syntheses that could be offered by commercial suppliers in China were quoted and calculated (Supplementary Table 5). As

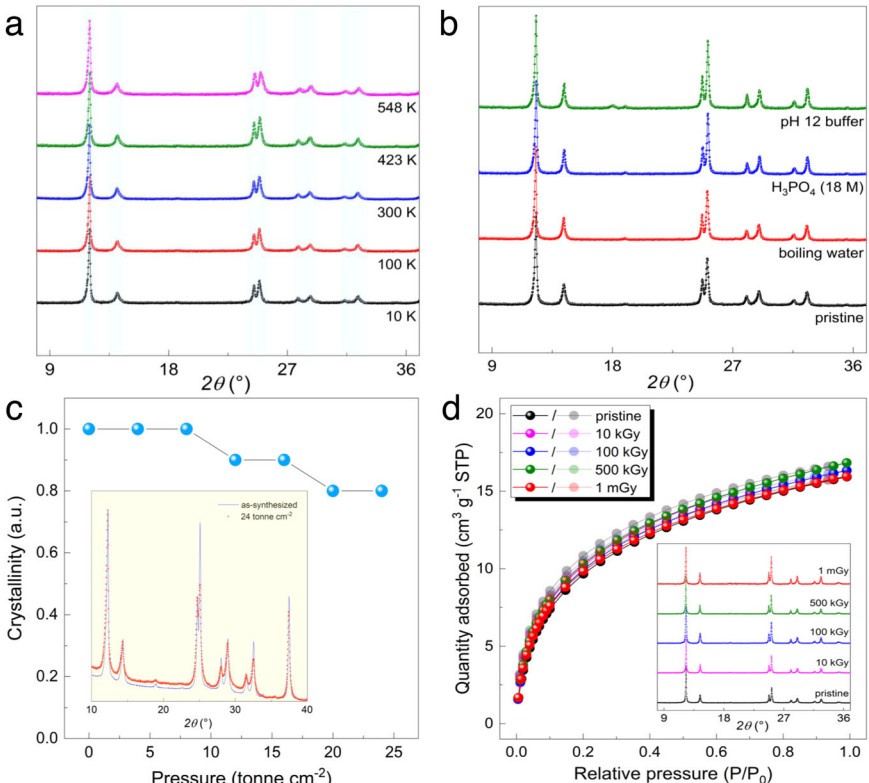

**Fig. 4 | Structural resistances of USTC-700 under various conditions.**
**a** Temperature-dependent PXRD patterns collected in a wide temperature range
(10 K to 548 K) showing a good thermal stability and a zero thermal expansion
property of the USTC-700 structure. **b** A good chemical stability of USTC-700
indicated by PXRD patterns collected on the samples treated under some pre-
viously unachievable conditions for Ti-framework compounds. **c** Mechanical sta-
bility reflected by the normalized crystallinity as a function of applied pressure on
the sample pellet. Inset, PXRD pattern comparison between the pristine sample and
the sample after compression at a pressure of 24 tonne cm$^{-2}$. **d** A good radiolytic
stability was supported by carbon dioxide ($CO_2$) sorption isotherms of USTC-700
samples exposed to gamma irradiation with various doses (fadeless symbols:
adsorption branches; transparent symbols: desorption branches). Inset, PXRD
patterns collected on samples correspond to each gamma irradiation dose.

shown in Fig. 3b, it is understandable that all the POCs and MOFs built
with complex organic linkers are costly, reaching the costs of around
100 GBP per gram or even more, as those chemicals involved in their
syntheses are not common feedstocks. Cost-effective sorbents that
include USTC-700 are all prepared by feedstock chemicals via one- or
two-step reactions, largely decreasing the overall price to less than 1
GBP per gram. Furthermore, mild and green synthesis condition with a
high STY is preferred for practical applications, which expects the
reaction to feature green chemicals, ambient pressure, low tempera-
ture, sufficient reactant concentration and short duration. However, it
is still challenging to combine all these features in the preparation of
one KQS material (Supplementary Note 5, Supplementary Tables 4-6).
It is noteworthy that USTC-700 was prepared in diluted HCl under
reflux condition at ambient pressure to afford a decent STY (Fig. 3b),
and its STY and unit cost are of the same level as those of benchmark
Ca-CHA and MgX zeolites, showing a great promise for practical
applications. Thus, USTC-700 fully addresses the green, safe and effi-
cient requirements in scale-up preparations.

Durability is as important as aforesaid characters towards prac-
tical usage for a material. The most straightforward property that
corresponds to durability is the stability of a material under certain
working condition, which usually covers thermal, chemical and
mechanical resistances. In particular, structural stability of the material
under irradiation for hydrogen isotope separation is required, since
the ultimate quantum sieves are expected to deal with the separation
and enrichment of radioactive tritium. To this end, we carried out
comprehensive tests and measurements to investigate the stability of
USTC-700, and the corresponding results are presented in Fig. 4.

Robustness of a crystalline material structure upon the external
temperature change, either cooling or heating, is mainly reflected by
its thermal resistance and unit-cell expansion/contraction behavior. In
general, the higher of the organic content, the weaker of the material
thermal stability in the presence of oxygen. Correspondingly, purely
organic POCs, COFs and carbon molecular sieves hardly keep their
structural integrities upon heating, while elevated thermal stabilities
are usually noticed for inorganic–organic hybrids as a result of the
increased ratio of inorganic content. The long-range order of the
USTC-700 structure could be maintained upon heating in air up to
300 °C, as confirmed by a combined analysis of temperature-
dependent PXRD and thermogravimetric data (Supplementary
Figs. 13 and 14). It is worth noting that a zero-thermal expansion of the
structure was observed in a wide temperature range of 10–548 K
(Fig. 4a, Supplementary Fig. 15 and Supplementary Table 7), which
possibly benefit the quantum sieving performance and could be
helpful to better understand the structure–property relationship.
Since quantum sieving performance highly depends on the pore
environment and working temperature, both positive and negative
thermal expansions of the material structures would lead to unpre-
dictable and incontrollable effects on the separation performance, as
the pore characters could be of notable differences between the gas
separation temperature (below the liquid $N_2$ temperature) and the
crystal data collection temperature (usually above 77 K). In contrast,
USTC-700 steadily preserves its pore size and shape with only local
bond flexibilities (rotating and stretching) under working conditions,
probably due to the high rigidity of both inorganic and organic com-
ponents, making it a predictable material for KQS.

Chemical stability, in particular water resistance, is gradually recognized to be an important issue of material for large-scale applications. Although the adsorption and separation of hydrogen isotopes are executed under anhydrous conditions, the scale-up material preparation, post-treatment, activation, and related handling processes are often carried out in air that generally contains a certain amount of water vapor. Moreover, it would be much more energy- and effort-consuming to upgrade the working condition to a water-free level. Therefore, comprehensive evaluation of the chemical stability of a material is indeed necessary before considering its feasibility towards practical applications. In this regard, USTC-700 displays a good chemical stability under various aqueous conditions, which is comparable to benchmark zeolites (Fig. 4b, Supplementary Figs. 16 and 17, and Supplementary Table 8). The crystalline structure of USTC-700 not only remains intact in boiling water for a long duration (more than 48 h), it also displays a good resistance to some harsh conditions that known Ti-based materials are not able to achieve yet, such as in concentrated $H_3PO_4$ and buffer with medium basicity. Hence, this exceptional chemical stability in all conditions ensures the durability and easy-handling of USTC-700 product in practical applications.

Mechanical stability of a sorbent is critical for shaping and packing processes in scale-up applications. Following the similar methods reported to evaluate those benchmarks for other applications[34,35], the mechanical stability of USTC-700 under pressure was assessed (Fig. 4c, Supplementary Note 1). It is evident that USTC-700 structure displays a high mechanical stability with nearly no crystallinity change even at the high pressure around 10 tonne $cm^{-2}$, at which MOFs present notable crystallinity decreases[34–37]. Along the increase of applied pressure up to 24 tonne $cm^{-2}$, the corresponding normalized crystallinity is still good enough to keep the structural long-range order (Supplementary Fig. 18), which makes USTC-700 standing among the most mechanically stable porous materials, to our knowledge, and is surely adequate for the shaping and packing processes in the practical applications.

It is known that both nonradioactive deuterium and radioactive tritium are essential for nuclear fusion to generate remarkable energy. The investigation of KQS material in the current stage is focusing on the separation of protium and deuterium, due to the apparent availability and safety of handling deuterium over tritium. It not only discovers promising sorbent to separate deuterium, but also accumulates fundamental knowledge and experimental experience to develop potential KQS material for the tritium separation and enrichment as the ultimate target. In this regard, a good radiolytic resistance turns to be an inevitable character in the optimization of quantum sieves for hydrogen isotope separation. While it is well documented that organic compounds are extremely sensitive to irradiation, recent advances have shown that metal-organic frameworks with weak bonding easily lose long-rang order or even short-range order to decrease porosity significantly under irradiation[37–40]. As a result, most organic-containing candidates shown in Fig. 3 are likely to have problems to keep their separation performances when they are exposed to radiation. To check the radiolytic stability of USTC-700, 300 mg-sized samples from the same batch were exposed to diverse gamma irradiation doses using a $^{60}$Co-$\gamma$ source. PXRD and $CO_2$ sorption measurements were carried out to characterize those post-irradiation samples. As shown in Fig. 4d, the $CO_2$ physisorption and PXRD data collected on the samples after gamma irradiation with diverse doses show negligible difference compared with those of the pristine sample, which suggests that USTC-700 maintains its structural integrity (crystallinity and pore character) well after exposure to gamma irradiation, even in the case of high dose of 1 mGy. It verifies that USTC-700 possesses adequate resistance in separating radioactive compounds, and it is among the most radiolytically stable metal-organic hybrid materials, to our knowledge.

Based on the fundamental foundation of the structure-performance relationship accumulated in previous research, we have designed and synthesized USTC-700 as a representative example of the next generation material for deuterium separation via KQS. A comprehensive assessment of the USTC-700's performance towards the requests of practical applications was carried out, addressing the separation selectivity and uptake, sorbent regeneration and cycling, cost, scalability and most importantly durability under working conditions. Computational simulations, such as semi-classical force field and quantum calculations, reveal that the good $D_2/H_2$ selectivity on USTC-700 is afforded by a unique interplay between diffusion kinetics, adsorption binding energy and nuclear quantum effects of the hydrogen isotopes under the strict confinement in the material porosity. This work not only provides a possible solution for hydrogen isotope separation towards industrial application, but also represents a leap forward in the field of isotope quantum sieving that developing practical materials starts to be of the same importance as fundamental research.

## Methods
### Synthesis of USTC-700
To a round bottom flask (25 mL), squaric acid (114 mg, 1 mmol) and diluted HCl (3 M, 10 mL) were added. The mixture was stirred at room temperature (RT) for dispersion. Titanium precursor (1 mmol) was added at last while stirring. The reaction mixture was refluxed at 120 °C for 24 h. After cooling down to RT, the crude product of USTC-700 was collected by filtration, washed with EtOH and air dry.

### Thermal desorption spectroscopy (TDS) for direct isotope separation
The selective adsorption after exposure to $D_2/H_2$ isotope mixtures was directly measured by the in-house designed setup of TDS. For a typical process, about 4 mg sample was loaded in the sample holder and activated at 393 K under vacuum for 5 h. The sample chamber is in an ultrahigh vacuum at RT, and cool down to aimed temperature at 30 K, 40 K, and 50 K, respectively. Then, the sample is exposed to a defined 1:1 $D_2/H_2$ mixture atmosphere (10, 100, and 300 mbar) for a fixed time (10, 60, and 120 min). After 1:1 $D_2/H_2$ mixture loading at the given exposure temperature and pressure, the gas molecules that had not been adsorbed were pumped out. Afterwards, the sample is rapidly cooled to the boiling temperature of the adsorbed gas. Finally, a linear heating ramp (0.1 K $s^{-1}$) is applied in order to thermally activate desorption. The desorbing gas is continuously detected using a quadrupole mass spectrometer, recognizing a pressure increase in the sample chamber when gas desorbs. The area under the desorption peak is proportional to the desorbing amount of gas.

## Data availability
All data involved in this work are included in this article and the corresponding supplementary materials. Crystal structure of USTC-700 is available at the CCDC database under CCDC-2175778.

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

## Acknowledgements

Q.Y., S.W., and J.D. acknowledge the fund support of the National Natural Science Foundation of China (22071234 and 21790350), the Chinese Academy of Sciences (XDC07000000), the Fundamental Research Funds for the Central Universities (WK9990000113 and WK2480000007), and the CAS Talent Introduction Program (Category B, KJ9990007009). P.G. acknowledges financial support from the National Natural Science Foundation of China (21972136). N.Y. acknowledges financial support from the National Natural Science Foundation of China (22102177) and the CAS Special Research Assistant Program. S.W. thanks Prof. Shan Jiang from Shanghai Tech University, Prof. Hailong Jiang and Dr. Yinhua Zhao from University of Science and Technology of China for their support and helpful discussion. The computational work was performed using HPC resources from GENCI-CINES (Grant A0120907613).

## Author contributions

Q.Y. contributed to the synthesis and general characterization of USTC-700, stability tests and contributed to the writing of the manuscript. J.W. N.Y., and P.G. contributed to the structure determination of USTC-700. L.Z. and M.H. contributed to the $H_2/D_2$ sorption and separation data collection and analysis, led the writing and revising of the manuscript. L.Y., J.L., and H.W. contributed to the general characterization of USTC-700 and the breakthrough sorption data collection and analysis. M.W., R.D., and G.M. performed molecular simulations, analyzed the results and contributed to writing and revising of the manuscript. S.W. and J.D. supervised the material synthesis, general characterization and stability tests, led the writing of the manuscript and coordinated the study.

## Competing interests

The authors declare no competing interests.
