## [Peer Review File · Nature Communications]

A squarate-pillared titanium oxide quantum sieve towards practical hydrogen isotope separationREVIEWER COMMENTS

Reviewer #1 (Remarks to the Author):

Yan et al report the hydrogen/deuterium (H₂/D₂) separation properties of a new organic-inorganic hybrid framework based on titanium oxide. The material, named USTC-700, consists of a rigid structure with pores of about ~3 Å in diameter. It absorbs both H₂ and D₂ gases, but at low temperature (typically 30K – 60K) it absorbs more D₂ than H₂. This is attributed to a phenomenon known as quantum sieving, whereby molecules with de Broglie wavelength larger than or comparable to the pore size of a material (in the angstrom range) can be excluded from absorbing in the pores of the material. The authors demonstrate that their material has a high D₂/H₂ selectivity (7.3-9.5). This comes with the trade-off of lower D₂ uptake than other materials, as expected. Where USTC-700 truly stands out is in its potential for use in real separation processes. The authors show that the mechanical stability (measured from the thermal contraction/expansion), chemical stability (as judged by water resistance) and, crucially cost, are significantly better than other microporous materials studied for this purpose. When all the parameters are considered together, the authors make a compelling case for the use of USTC-700 in real applications. On this basis, I would like to recommend the manuscript for publication.

It would be helpful for the authors to elaborate on how they made the comparison of selectivity and D₂ uptake with other materials. As they demonstrate in the manuscript, their material's selectivity changes from 7.3-9.5 depending on temperature and exposure time (although not pressure). This suggests that the data point in fig. 3a (red star) should have an error bar/shaded region unless all materials are compared at the same temperature and exposure time. Whichever the case, this should not change the conclusions of the manuscript and a couple lines in the SI/figure caption should suffice.

Reviewer #2 (Remarks to the Author):

This work reports a MOF quantum sieve for hydrogen isotope separation. A new ultra-microporous squarate pillared titanium oxide hybrid framework (USTC-700) was synthesized via a cost-effective green synthesis approach using chemical feedstock as raw material. The MOF material exhibited high D₂/H₂ selectivity and good D₂ uptake, and an outstanding space-time yield with low cost. This study is comprehensive and well-organized. Although the MOF structure is not new, the finding of facile approach to synthesize Ti-based MOFs and excellent stability under working conditions presents a notable step towards gas isotopes separation based on quantum sieving materials. I would recommend its publication in Nature Communications after addressing the following points:

1. The authors are suggested to compare their USTC-700 with its analogues such as UTSA-280 molecular sieves that were also studied for gas separation (e.g., AIChE Journal, 2022, 68, e17688) in terms of synthesis and structure.
2. Why the thermal stability at high temperature is important for this material? Because the D₂/H₂ separation is conducted at very low temperature.
3. How about the cycling performance of the column breakthrough measurements? This is a very important parameter for sorption application.
4. The authors showed outstanding CO₂ adsorption property after chemical stability test under harsh conditions. How about the D₂/H₂ separation performance after the stability test?

Reviewer #3 (Remarks to the Author):

In this manuscript "A squarate-pillared titanium oxide quantum sieve towards practical hydrogen isotope separation", the authors reported a novel Ti-squarate framework (USTC-700) and its separation for D₂/H₂. The sample USTC-700 takes lots of advantages in thermal and chemical stabilities and scale-up. However, the D₂/H₂ KQS separation performance prevents its publication in journal Nature Communications. The comments are listed as below:

- 1) In Fig.2a, obvious lags were observed in desorption isotherms. Did it be caused by not balancing of adsorption or by impurities in the gas? For each pressure steps, the adsorption/desorption kinetic data should be collected and analyzed when discussing kinetic quantum sieving. If such data could not be obtained, at least the duration time to reach equilibrium for each step should be provided.
- 2) At 1 mbar, the D2 uptake was 15 % higher than that of H2 at 30 K due to the TQS effect. However, Fig.2c showed a very large difference of TDS pecks for D2/H2. The authors explained the reason was fast diffusion kinetics for D2 than H2 in the initial 10 minutes. But the inset in Fig.2d showed the selectivity increased with longer exposure time at 30 K, from 7.3 to 9.5 after 120 min exposure. The authors thought that the H2 molecules were replaced gradually by the adsorbed D2. It was unreasonable. If the authors exposed the sample in cryogenic temperature for an enough longer time to get complete adsorption equilibrium on USTC-700 for a D2/H2 mixture and then test TDS, what could be happened? It could be self-contradiction with the results reported in this work.
- 3) If, as the authors claimed, there was a significant KQS effect, the peak location of D2 should be significantly earlier than that of H2 at low temperature. However, as showed in Fig.3c, the two peaks occurred almost at the same time. The only larger TDS peak area for D2 on sample USTC-700 did not prove the existence of KQS effect.
- 4) One question for TDS tests. "After 1:1 D2/H2 mixture loading at the given exposure temperature and pressure, the gas molecules that had not been adsorbed were pumped out." A certain amount of the guest molecules adsorbed in the sample could be inevitably pumped out in the meantime. H2 is easier to escape from the pore, considering the slightly weaker interaction between the framework and H2, thus, the TDS method used in this work might lead to an overestimated selectivity of D2/H2.
- 5) For the separation of hydrogen isotope mixtures with porous materials, the capacity and selectivity at 77 K (liquid N2 temperature) are considered to be the essential parameters for the promising application. However, USTC-700 has both poor adsorption capacity and selectivity at 77 K. So it is not a practical material for D2/H2 KQS or TQS separation.
- 6) In Fig.2c, there are two distinct peaks of thermal desorption spectra for 30 K and 40 K, but only one for 50 K. What makes the differences between them?
- 7) In Fig.3 and Supplementary Table 4, the conditions for selectivity and uptakes of each sorbent listed should be noted.
- 8) In Supplementary Fig.5, the heat of adsorption of D2 and H2 increases with the increasing of surface coverage, which is unusual and should be explained.
- 9) In Supplementary Fig.6, the authors mentioned "Thermal desorption spectra (TDS) of USTC-700 obtained after exposure to a 10 mbar 1:1 D2/H2 isotope mixture at 30 K for an exposure time from 10 to 300 min", but no TDS of USTC-700 exceed 120 min was provided.
- 10) It is suggested to put the sorption isotherms of H2 and D2 at the same temperature in one figure for comparison purpose.
- 11) The title of Reference 4 is not correct. It should be "Nuclear power can help the democratic world achieve energy independence".

RESPONSE TO REVIEWERS' COMMENTS

Reviewer #1 (Remarks to the Author):

Yan et al report the hydrogen/deuterium (H_2/D_2) separation properties of a new organic-inorganic hybrid framework based on titanium oxide. The material, named USTC-700, consists of a rigid structure with pores of about ~ 3 Å in diameter. It absorbs both H_2 and D_2 gases, but at low temperature (typically 30 K – 60 K) it absorbs more D_2 than H_2 . This is attributed to a phenomenon known as quantum sieving, whereby molecules with de Broglie wavelength larger than or comparable to the pore size of a material (in the angstrom range) can be excluded from absorbing in the pores of the material. The authors demonstrate that their material has a high D_2/H_2 selectivity (7.3-9.5). This comes with the trade-off of lower D_2 uptake than other materials, as expected. Where USTC-700 truly stands out is in its potential for use in real separation processes. The authors show that the mechanical stability (measured from the thermal contraction/expansion), chemical stability (as judged by water resistance) and, crucially cost, are significantly better than other microporous materials studied for this purpose. When all the parameters are considered together, the authors make a compelling case for the use of USTC-700 in real applications. On this basis, I would like to recommend the manuscript for publication.

We thank the referee for the support and positive evaluation of our work.

Question 1: It would be helpful for the authors to elaborate on how they made the comparison of selectivity and D_2 uptake with other materials. As they demonstrate in the manuscript, their material's selectivity changes from 7.3-9.5 depending on temperature and exposure time (although not pressure). This suggests that the data point in fig. 3a (red star) should have an error bar/shaded region unless all materials are compared at the same temperature and exposure time. Whichever the case, this should not change the conclusions of the manuscript and a couple lines in the SI/figure caption should suffice.

Answer: We thank the referee for this suggestion. In Fig. 3 (main text) and Supplementary Table 4, all the selectivity and D_2 uptake data listed for comparison correspond to the best performance of each material, including the case of USTC-700 (red star). We have added the related condition, including measurement temperature and pressure, of each material in the revised manuscript and supplementary information accordingly.

Reviewer #2 (Remarks to the Author):

This work reports a MOF quantum sieve for hydrogen isotope separation. A new ultra-microporous squarate pillared titanium oxide hybrid framework (USTC-700) was synthesized via a cost-effective green synthesis approach using chemical feedstock as raw material. The MOF material exhibited high D_2/H_2 selectivity and good D_2 uptake, and an outstanding space-time yield with low cost. This study is comprehensive and well-organized. Although the MOF structure is not new, the finding of facile approach

to synthesize Ti-based MOFs and excellent stability under working conditions presents a notable step towards gas isotopes separation based on quantum sieving materials. I would recommend its publication in Nature Communications after addressing the following points.

We thank the referee for the support and positive evaluation of our work.

We apologize for the lack of clarity in addressing the structure novelty of USTC-700 in the manuscript, which may have given the impression that the structure is already known and we had developed an alternative approach to synthesize it. To avoid the aforementioned ambiguity, we have clearly pointed out that USTC-700 is of a new structure in the revised manuscript.

Question 1: The authors are suggested to compare their USTC-700 with its analogues such as UTSA-280 molecular sieves that were also studied for gas separation (e.g., AIChE Journal, 2022, 68, e17688) in terms of synthesis and structure.

Answer: As we mentioned in our previous response, we apologize for not addressing the structure novelty of USTC-700 in the earlier manuscript. Though UTSA-280 and USTC-700 are of different crystal structures, it is interesting to compare them since they share some important structural characters, such as infinite inorganic building units and ultramicropore systems. UTSA-280 molecular sieve is of a three-dimensional (3D) structure built with one-dimensional (1D) calcium-oxo chain inorganic building unit and squarate linker. The differences of structure information between UTSA-280 and USTC-700 are summarized in Table R1, and the related details are shown in Figure R1.

Table R1. Structure information of UTSA-280 and USTC-700.

Material	UTSA-280	USTC-700
Space Group	tetragonal I-42d	monoclinic P2/m
Unit Cell Parameters	$a = b = 13.615(2) \text{ \AA}$ $c = 7.675(1) \text{ \AA}$ $V = 1422.7 \text{ \AA}^3$	$a = 3.6974(2) \text{ \AA}$ $b = 7.1931(5) \text{ \AA}$ $c = 6.4272(5) \text{ \AA}$ $\beta = 106.633(4)^\circ$ $V = 163.78(2) \text{ \AA}^3$
Inorganic Building Unit	1D Ca-oxo chain	2D $(\text{Ti}_2\text{O}_3)_n$ layer
Squarate Function	spacer to connect Ca-oxo chains	pillar to connect $(\text{Ti}_2\text{O}_3)_n$ layers
Pore Diameter	$3.2 \times 4.5 \text{ \AA}^2$ $3.8 \times 3.8 \text{ \AA}^2$	$3.1 \times 3.1 \text{ \AA}^2$

Figure R1. Structure information of UTSA-280. **a)** The perpendicular arrangement of 1D calcium-oxo chains in the crystal structure. **b)** Adjacent chain SBUs are interconnected by squarate linker molecules, generating the 3D network viewed along the *c*-axis. **c)** Structure viewed along the *b*-axis. **d)** PXRD pattern and unit cell parameter comparison between UTSA-280 and USTC-700.

As reported in the literature (*Nat. Mater.*, 2018, 17, 1128-1133), the synthesis of UTSA-280 is green, scalable and cost-effective, which is similar to that of USTC-700. UTSA-280 was reported to hold a great promise in molecular sieving, including C₂H₄/C₂H₆ separation (*Nat. Mater.*, 2018, 17, 1128-1133; *AIChE J.*, 2022, 68, e17688), water/ethanol separation (*Angew. Chem. Int. Ed.*, 2023, 62, e202216710), and Xe/Kr separation (*J. Phys. Chem. C.*, 2020, 124, 14603-14612), to name a few. Therefore, both UTSA-280 and USTC-700 are good molecular sieving candidates towards practical separations.

Question 2: Why the thermal stability at high temperature is important for this material? Because the D₂/H₂ separation is conducted at very low temperature.

Answer: It is true that the D₂/H₂ separation is conducted at low temperatures (below 77 K in general). However, it is necessary to activate the sorbent to make the structural

pore as accessible as possible before the material is subject to separation. Thermal activation that heats porous sorbent at higher temperatures is one of the most common methods, which requires a certain thermal stability for a given material. Otherwise, much more time- and effort-consuming activation methods, such as low boiling point solvent exchange and supercritical CO₂ activation, are needed for thermally instable materials. In the case of USTC-700, we found that the heating temperature above 150 °C with a duration longer than six hours is crucial for its activation. Thus, thermal stability at high temperature is important for the material activation before the D₂/H₂ separation.

Question 3: How about the cycling performance of the column breakthrough measurements? This is a very important parameter for sorption application.

Answer: We thank the referee for this suggestion. Accordingly, we carried out column breakthrough measurements at 77 K for the mixed gases of H₂/D₂/Ne (1/1/98, vol. %) for three cycling runs. The corresponding results are shown in Figure R2 and Supplementary Figure 12 in the revised supplementary information. USTC-700 displays a good cycling performance in column breakthrough measurements.

Figure R2. Cycling performance of USTC-700 in column breakthrough measurements at 77 K for the mixed gases of H₂/D₂/Ne (1/1/98 vol.%). **a)** The dynamic breakthrough curves of three cycling runs. **b)** Column breakthrough curves of three cycling runs.

Question 4: The authors showed outstanding CO₂ adsorption property after chemical stability test under harsh conditions. How about the D₂/H₂ separation performance after the stability test?

Answer: We thank the referee for this suggestion. Accordingly, we measured the D₂/H₂ separation by TDS on the USTC-700 samples after chemical treatments (concentrated HNO₃ and boiling water), and the performance shows high reproducibility. The position

of the desorption peaks and total uptake of isotopes and D₂/H₂ selectivity are almost identical. The corresponding results are shown in Table R2 and Figure R3. We have also included these results as Supplementary Table 8 and Supplementary Figure 17 in the revised supplementary information.

Table R2. Summary of TDS measurement data on USTC-700 samples after chemical treatments.

Sample	Total Uptake (mmol/g)	Selectivity
Pristine	1.07	7.3
HNO ₃ -treated	1.01	7.5
Boiling Water-treated	1.10	7.5

Figure R3. TDS results of USTC-700 samples after chemical treatments (please note that the intense of each peak varies, since each batch of sample are not 100% homogenous, and TDS is extremely sensitive to minor variation).

Reviewer #3 (Remarks to the Author):

In this manuscript “A squarate-pillared titanium oxide quantum sieve towards practical hydrogen isotope separation”, the authors reported a novel Ti-squarate framework (USTC-700) and its separation for D₂/H₂. The sample USTC-700 takes lots of advantages in thermal and chemical stabilities and scale-up. However, the D₂/H₂ KQS separation performance prevents its publication in journal Nature Communications. The comments are listed as below:

Question 1: In Fig.2a, obvious lags were observed in desorption isotherms. Did it be caused by not balancing of adsorption or by impurities in the gas? For each pressure steps, the adsorption/desorption kinetic data should be collected and analyzed when discussing kinetic quantum sieving. If such data could not be obtained, at least the duration time to reach equilibrium for each step should be provided.

Answer: We thank the referee to point this out. Unfortunately, it is not easy for us to obtain the kinetic data directly. The hysteresis is caused by the slow diffusion in adsorption versus desorption, considering USTC-700 possesses very small pores. For each step the equilibrium time varies since the instrument reads the pressure to reach equilibrium but not the time. It takes ~400 min for each measurement, including adsorption and desorption. The following is a table to show the equilibrium time of each isotherm for H₂ and D₂ at different temperatures (Table R3). At 30 K, the equilibrium time of H₂ sorption is much longer than that of D₂. Noted that KQS generally occurs in the very first steps of the isotherms and at low temperature, therefore at temperatures higher than 30 K, it is difficult to observe KQS directly from the full single component sorption isotherms.

Table R3. Summary of equilibrium time of H₂ and D₂ sorption at different temperatures.

Temperature/K	Equilibrium time/min	
	H ₂	D ₂
30	586.9	464.8
40	339.3	449.2
50	378.0	414.1
60	363.8	427.2
70	379.2	382.1
77	373.9	338.6

Question 2: At 1 mbar, the D₂ uptake was 15% higher than that of H₂ at 30 K due to the TQS effect. However, Fig. 2c showed a very large difference of TDS peaks for D₂/H₂. The authors explained the reason was fast diffusion kinetics for D₂ than H₂ in the initial 10 minutes. But the inset in Fig. 2d showed the selectivity increased with longer exposure time at 30 K, from 7.3 to 9.5 after 120 min exposure. The authors thought that the H₂ molecules were replaced gradually by the adsorbed D₂. It was unreasonable. If the authors exposed the sample in cryogenic temperature for an enough longer time to get complete adsorption equilibrium on USTC-700 for a D₂/H₂ mixture and then test TDS, what could be happened? It could be self-contradiction with the results reported in this work.

Answer: Indeed, based on H₂ and D₂ isotherms measured at 30 K, D₂ uptake is 15% higher than that of H₂. It is generally observed in nanoporous materials that they adsorb more D₂ than H₂. And these results are based on the single component pure gas

adsorption measurements. However, as the referee pointed out, a very large difference of TDS peaks for D₂/H₂, this ascribes to a competitive measurement carried out directly on a mixture of H₂ and D₂, which is different from single component gas adsorption measurement.

To make our point more understandable, we would like to specify some important information about TDS measurement. Generally, thermal desorption spectra reflect the state created during the gas adsorption at a given exposure temperature and exposure time. Fig. 2c shows the desorption spectra of H₂ and D₂ after USTC-700 being exposed to a 1:1 H₂/D₂ mixture at 10 mbar for 10 minutes at temperatures of 30, 40, and 50 K; the exposure temperatures are indicated in the upper right corner of each figure. After being thus exposed, the sample chamber is evacuated directly at the exposure temperature for removing unadsorbed gas molecules, prior to cooling down to 20 K. The sample is then heated with a constant heating rate under vacuum, and the desorbing gases are monitored with a quadrupole mass spectrometer. For porous materials with large pore apertures, there is no diffusion limitation for the penetrating gas, and hence all adsorption sites are accessible during gas exposure. The desorption spectra can then reflect directly the distribution of different adsorption energies. By contrast, for materials with small pore apertures, the adsorption process is limited by the diffusion-limited process of gas penetration into the material.

We agree to the referee's point that the quantum effect may not be very evident in normal conditions. However, owing to such structure of USTC-700 possessing very narrow aperture of around 3 Å, the larger adsorption of D₂ than that of H₂ can be observed. Under the identical condition (P, t, T), hence, USTC-700 can generally adsorb more D₂ than H₂ due to the fast kinetics and more favorable sorption affinity caused by electrostatic effect from the functional groups (thermodynamic). When the combination effect occurs, longer exposure time can lead to a favored adsorption for D₂, though it's not the major effect compared to KQS. Please also note that the high D₂ uptake compared to H₂ in porous materials is generally observed below 77 K, but this trend can be also shifted to even over 77 K by providing strong binding sites (in NaA zeolite, *Langmuir*, 1998, 14, 7255-7259; in Cu-MFU-4l, *Nat. Comm.*, 2017, 8, 14496). Here, no adsorption peak observed above 77 K, therefore, we believe this larger adsorption of D₂ mainly be originated to the effect of kinetics.

Last but not least, the referee asked what would happen if we expose the sample to isotopic mixture long enough. However, based on the experimental procedure, TDS is not eligible for equilibrium measurement, since the un-adsorbed molecules have to be removed anyway, this leads to the system not under equilibrium condition.

Question 3: If, as the authors claimed, there was a significant KQS effect, the peak location of D₂ should be significantly earlier than that of H₂ at low temperature.

However, as showed in Fig. 2c, the two peaks occurred almost at the same time. The only larger TDS peak area for D₂ on sample USTC-700 did not prove the existence of KQS effect.

Answer: We highly appreciate that the referee brought this issue to our notice. We also agree that the desorption peak for D₂ could be earlier than that of H₂ under ideal conditions. However, the real condition is distinct. Firstly, the KQS is theoretically proposed in a split cylindrical pore. But for the real materials, it is impossible to achieve this, since the electrostatic effect exists due to the ligand, pillar, metal etc. Secondly, the measured desorption rate of a TDS spectrum is determined by the initial coverage, the rate constant, and the desorption order, making the determination of the positions of desorption peaks complicated. For our system, the identical desorption maxima have been observed. The same phenomena have also been reported for KQS in many other systems, for example, porous organic cages (*Science*, 2019, 366, 613-620), metal-amide-imidazolate frameworks with narrow 1D channels (*ChemPhysChem*, 2019, 20, 1311-1315) etc. Therefore, we believe this larger adsorption of D₂, at the identical maximum temperature, is originated in the KQS effect.

Question 4: One question for TDS tests. “After 1:1 D₂/H₂ mixture loading at the given exposure temperature and pressure, the gas molecules that had not been adsorbed were pumped out.” A certain amount of the guest molecules adsorbed in the sample could be inevitably pumped out in the meantime. H₂ is easier to escape from the pore, considering the slightly weaker interaction between the framework and H₂, thus, the TDS method used in this work might lead to an overestimated selectivity of D₂/H₂.

Answer: The density of the particle in the pores or channels is dependent on the energy level of the molecule, and therefore the mass (*Chem. Phys. Lett.*, 1995, 232, 379-382). Owing to the higher mass, energy of deuterium is lower and therefore its density in the channel will be increased. As discussed, USTC-700 possesses a narrow pore aperture, which is comparable to the kinetic diameter of hydrogen molecules. Therefore, the separation of hydrogen isotopes can only take place close to the entrance of the pore opening, based on the density of the molecules, since after penetrating the narrow pore channels, no passing of gas molecules is possible, which is called single-file filling (*Angew. Chem. Int. Ed.*, 2022, 61, e202202450). Under this condition, the molecules will be removed, if any, in the same sequence as when they are filling in. Logically, we will lose more D₂ than H₂, if any loss of molecules happened. Thus, the selectivity might be underestimated, but not overestimated.

Question 5: For the separation of hydrogen isotope mixtures with porous materials, the capacity and selectivity at 77 K (liquid N₂ temperature) are considered to be the essential parameters for the promising application. However, USTC-700 has both poor adsorption capacity and selectivity at 77 K. So it is not a practical material for D₂/H₂

KQS or TQS separation.

Answer: The technological processes used currently for hydrogen isotope separation are cryogenic distillation operated at 20 to 25 K or the Girdler method via H₂S cycle and heavy water electrolysis. Both have a selectivity below 2.5 and are highly energy and cost intensive. Quantum sieving proposed here shows a selectivity of 9.5 at 30 K, which is superior and less energy intensive. The quantum sieving effect is strongly temperature dependent and the selectivity drops with increasing temperature. In a review, Kowalczyk et al. summarized measurements in carbons and zeolites performed at 77 K and 87 K at low coverage and they all show a selectivity below 3. In extensive path integral Monte Carlo simulations, they showed the strong temperature dependence between 77 K and 33 K. In addition, though chemical affinity sieving (CAS) via existence of open metal sites can enhance the separation temperature above liquid nitrogen temperature, which appears advantageous over KQS, the stability of the active sites under harsh, or even technical conditions still remains a challenge.

Moreover, as for the practicality of operation at 30 K, this is common in the Fusion projects. For example, JET (Joint European Torus) still using cryogenic distillation at scale and at temperatures < 30 K, so for a fusion reactor application, at least, it is practical to run this separation at 30 K (N. Bainbridge et al., *Fusion Eng. Des.*, 1999, 47, 321–332). We would also assert that a 30 K adsorption process is likely to be better than cryogenic distillation (operation between 20 to 25 K) in terms of energy cost.

Question 6: In Fig. 2c, there are two distinct peaks of thermal desorption spectra for 30 K and 40 K, but only one for 50 K. What makes the differences between them?

Answer: As response for Question 2, TDS spectra reflect the state created during the gas adsorption at a given exposure temperature and exposure time. Therefore, during desorption, the activation energy determines the desorption temperature at which the gas can leave the pores. Hence, in the case of 30 K, the TDS spectrum results suggest that there are two adsorption sites possessing different energy. With increasing exposure temperature, the desorption peaks in TDS spectra are shifted to higher temperatures and only one desorption peak left. This is because the weakly bond gas molecules have been removed already during evacuation at the exposure temperature.

Question 7: In Fig. 3 and Supplementary Table 4, the conditions for selectivity and uptakes of each sorbent listed should be noted.

Answer: We thank the referee for this suggestion. In Fig. 3 (main text) and Supplementary Table 4, all the selectivity and D₂ uptake data listed for comparison

correspond to the best performance of each material. We have added the related condition, including measurement temperature and pressure, of each material in the revised manuscript and supplementary information accordingly.

Question 8: In Supplementary Fig. 5, the heat of adsorption of D₂ and H₂ increases with the increasing of surface coverage, which is unusual and should be explained.

Answer: The isosteric heat of adsorption is calculated using a variant of the Clausius-Clapeyron equation which is based on the difference in pressure needed to reach a certain uptake at various temperatures. Therefore, the method is especially sensitive to experimental uncertainties in very steep and flat regions of the isotherm. To avoid this, only the intermediate region is used where the relation of uptake to pressure is more precise. The lack of this method is that there is no heat of adsorption for low surface coverage. This could be calculated with higher accuracy when doing a low pressure (up to 0.1 MPa) adsorption measurement, but then the high surface coverage heat of adsorption cannot be calculated. Therefore, the heat of adsorption for USTC-700 seems to be increasing but can be regarded as constant within the experimental error.

In addition, though generally the adsorption enthalpy decreases as loading increases, because the adsorbate molecules are firstly adsorbed at the strongest site and finally the weakest site, the adsorption enthalpy may sometimes increase as the loading increases because of increased adsorbate-adsorbate interaction and/or structural transformation of the adsorbent. This phenomenon has also been observed in other nanoporous materials for H₂ adsorption, for example, *Chem. Sci.*, 2017, 8, 7560-7565.

Question 9: In Supplementary Fig.6, the authors mentioned “Thermal desorption spectra (TDS) of USTC-700 obtained after exposure to a 10 mbar 1:1 D₂/H₂ isotope mixture at 30 K for an exposure time from 10 to 300 min”, but no TDS of USTC-700 exceed 120 min was provided.

Answer: We thank the referee for pointing this out. We are sorry to put 300 min by mistake. We have made the revision in the revised supplementary information.

Question 10: It is suggested to put the sorption isotherms of H₂ and D₂ at the same temperature in one figure for comparison purpose.

Answer: As suggested by the referee, we replotted the sorption isotherms of H₂ and D₂ at the same temperature in one figure, as shown in Figure R4. It is true that for each

temperature, the comparison between H₂ and D₂ sorption is more obvious. However, the decreases of H₂ and D₂ uptakes along the increase of data collection temperature are less clear. Moreover, it is much more space-consuming in comparison with the previous figure. Thus, we think it would be better to keep the previous Fig. 2a and b in the main text, and add Figure R4 in the revised supplementary information (Supplementary Figure 5) for a better comparison of H₂ and D₂ single component sorption at the same temperature.

Figure R4. H₂ and D₂ single component sorption isotherms collected at 30 K, 40 K, 50 K, 60 K, 70 K and 77 K, respectively.

Question: The title of Reference 4 is not correct. It should be “Nuclear power can help the democratic world achieve energy independence”.

Answer: We thank the referee for pointing out this. However, this problem is due to the inconsistency between the web-page and the PDF version for citation of this reference,

as shown below. In the published official PDF version that can be cited, the title is ‘Use nuclear power to end reliance on Russian oil’. It is also presented in Web of Science with this title. Nevertheless, the web-page of this paper shows the title of ‘Nuclear power can help the democratic world achieve energy independence’.

A personal take on science and society

World view

By Nicolas Mazzucchi

Use nuclear power to end reliance on Russian oil

Russia's invasion of Ukraine has highlighted the need to improve energy security.

Last month, after weeks of negotiations, European Union leaders agreed to ban 90% of Russian oil imports by 2023. Until then, Russia will be able to continue to sell millions of barrels of oil a day to the EU, with some of the proceeds continuing to fund the war. Reliance on this fuel delayed a dignified, united condemnation of the invasion of Ukraine, and continues to interfere with the EU's response.

“New nuclear technologies are more practical and more agile.”

fundamental, both in terms of the physics and the change it could represent for the industry. Fast neutron reactors operate with enough energy to cause fission of many heavy atoms, potentially eliminating both nuclear waste material and reliance on uranium as the sole fuel source. These are just one of a host of fourth-generation nuclear reactor systems that together overcome some of the shortcomings of conventional installations.

Russia and China are currently alone in operating commercial power plants using these technologies – at China's Shidaowan power plant in Shandong, and Beloyarsk-3 and -4 in Sverdlovsk Oblast, Russia.

nature

View all journals Search Log in

Explore content About the journal Publish with us Subscribe

Sign up for alerts RSS feed

nature > world view > article

WORLD VIEW | 28 June 2022

Nuclear power can help the democratic world achieve energy independence

Russia's invasion of Ukraine has highlighted the need to improve energy security.

Nicolas Mazzucchi

Last month, after weeks of negotiations, European Union leaders agreed to ban 90% of Russian oil imports by 2023. Until then, Russia will be able to continue to sell millions of barrels of oil a day to the EU, with some of the proceeds continuing to fund the war. Reliance on this fuel delayed a dignified, united condemnation of the invasion of Ukraine, and continues

You have full access to this article via University of Science and Technology of China

Download PDF

Related Articles

How a small nuclear war would transform the entire planet

Ukrainian researchers pressure journals to boycott Russian authors

How three Ukrainian scientists are surviving Russia's brutal war

REVIEWER COMMENTS

Reviewer #1 (Remarks to the Author):

The authors have answered the comments satisfactorily. I am happy to recommend the manuscript for publication.

Reviewer #2 (Remarks to the Author):

The authors have addressed the reviewers' comments and the revised manuscript is suggested to be published in Nature Communications.

Reviewer #3 (Remarks to the Author):

This manuscript have been revised by the authors, but the questions raised by this reviewer in the first round reviewing have been still existed:

1)The authors didn't provide the adsorption/desorption kinetic data, so this reviewer can not judge how fast for D2 than H2 on USTC-700. The authors said that it takes about 400 min for each measurement, including adsorption and desorption. From the figure 2a and 2b, each measurement includes about 50 pressure steps, so it only needs about 8 min to reach adsorption equilibrium. By this reviewer's experience of measuring hundreds H2/D2 isotherms, the H2/D2 kinetics were not very slow, actually quite fast! Therefore the hysteresis might be caused by both of impurities in the gas and/or un-equilibrium. In manuscript, the authors did not provide the grade D2 gas they used in this study. Usually, the purity of D2 in gas cylinder is 99.8% , much lower than H2 gas (99.999%). If the authors did not further purify D2 gas and used D2 cylinder gas directly, the D2 data obtained could be questionable.

2)We have known that it only needs ~ 8 min or slight longer time to reach H2/D2 adsorption equilibrium for 30K, so we can conclude that the pores of USTC-700 were fully filled by D2 and H2 after being exposed to a 1:1 H2/D2 mixture at 10 mbar for 10 minutes at temperatures of 30 K. This reviewer didn't comment the KQS effect at 30 K. The raised question was that why the selectivity increased with longer exposure time. The authors thought that H2 molecules were replaced gradually by the adsorbed D2. The D2/H2 from TDS was 7.3 for 10 min exposure, which was already much higher than the ratio of D2/H2 measured by isotherms. If the D2/H2 by TDS increased to 9.5 for 120 min exposure time, what was the driving force for it? The authors didn't answer the question reasonably.

3)This reviewer have had concerns about the TDS test and thought TDS method used in this work might lead to an overestimated selectivity of D2/H2. As the adsorption/desorption kinetics were not very slow, the operation of gas molecules pump out could lead to a more serious problem for overestimated selectivity of D2/H2.

4)The authors have not explained the question properly that why there are two distinct peaks of TDS for 30 K and 40 K, but only one for 50 K. Is it from D2 gas impurities?

RESPONSE TO REVIEWERS' COMMENTS

Reviewer #3 (Remarks to the Author):

This manuscript has been revised by the authors, but the questions raised by this reviewer in the first-round reviewing have been still existed:

Question 1: The authors didn't provide the adsorption/desorption kinetic data, so this reviewer can not judge how fast for D₂ than H₂ on USTC-700. The authors said that it takes about 400 min for each measurement, including adsorption and desorption. From the figure 2a and 2b, each measurement includes about 50 pressure steps, so it only needs about 8 min to reach adsorption equilibrium. By this reviewer's experience of measuring hundreds H₂/D₂ isotherms, the H₂/D₂ kinetics were not very slow, actually quite fast! Therefore, the hysteresis might be caused by both of impurities in the gas and/or un-equilibrium. In manuscript, the authors did not provide the grade D₂ gas they used in this study. Usually, the purity of D₂ in gas cylinder is 99.8%, much lower than H₂ gas (99.999%). If the authors did not further purify D₂ gas and used D₂ cylinder gas directly, the D₂ data obtained could be questionable.

Answer: As mentioned by the referee, the purity of the D₂ gas used in our measurements is 99.8%, which is consistent with the purity level of most commercially available gas cylinders. The purity of H₂ gas used for measurements is 99.999%.

For the collection of sorption isotherm data, we utilize gas cylinders. However, in nearly 100 measurements, we have not encountered any issues or indications of impurities, regardless of the presence or absence of hysteresis.

Based on existing literature and the fundamental principles of sorption, equilibrium at low pressures generally takes longer to establish compared to steps at higher pressures. Therefore, the calculation of the average equilibrium time for each step as 8 minutes is inaccurate. To illustrate this point, Table R1 presents a set of raw data collected for a D₂ sorption at 30 K. It is important to note that the initial several steps ($P < 1$ torr) required nearly 5 hours, clearly demonstrating the presence of diffusion barriers.

Table R1. Measurement details of each data point for D₂ sorption at 30 K.

Pressure Torr	P ₀ Torr	Volume @ STP cc	Time Minute
6.02662e-05	760.00	0.000313558	14.8
0.00187595	760.00	0.0301033	51.3
0.00394266	760.00	0.0978732	100.0
0.00771651	760.00	0.301779	180.6
0.0357277	760.00	0.961083	213.6
0.0803208	760.00	1.25974	236.7
0.252916	760.00	1.48723	252.9
0.42606	760.00	1.5603	267.2
0.551128	760.00	1.5976	281.6
0.881426	760.00	1.65762	290.5
2.44478	760.00	1.77946	303.7
3.99011	760.00	1.8408	318.7
5.60158	760.00	1.89187	325.9
7.67733	760.00	1.93426	334.2
23.5947	760.00	2.10882	338.1
40.4364	760.00	2.22919	341.9
56.3201	760.00	2.32288	345.6
78.9034	760.00	2.42866	349.1
156.07	760.00	2.79123	353.2
232.948	760.00	3.10248	357.4
309.335	760.00	3.39297	362.1
377.14	760.00	3.63909	365.8
461.096	760.00	3.93771	370.0
537.242	760.00	4.22218	374.5
605.35	760.00	4.45255	378.3
688.682	760.00	4.75823	382.8
753.72	760.00	4.98685	386.6
749.228	760.00	4.97913	390.4
686.214	760.00	4.75861	394.6
604.202	760.00	4.47291	399.9
527.845	760.00	4.20504	404.1
451.803	760.00	3.93572	408.4
375.168	760.00	3.68699	412.7
300.099	760.00	3.39349	416.9
224.458	760.00	3.10782	421.1
148.94	760.00	2.79147	425.3
73.584	760.00	2.46247	429.8
37.5664	760.00	2.29056	433.8
7.21519	760.00	1.98391	442.1

Question 2: We have known that it only needs ~ 8 min or slight longer time to reach H₂/D₂ adsorption equilibrium for 30 K, so we can conclude that the pores of USTC-700 were fully filled by D₂ and H₂ after being exposed to a 1:1 H₂/D₂ mixture at 10 mbar for 10 minutes at temperatures of 30 K. This reviewer didn't comment the KQS effect at 30 K. The raised question was that why the selectivity increased with longer exposure time. The authors thought that H₂ molecules were replaced gradually by the adsorbed D₂. The D₂/H₂ from TDS was 7.3 for 10 min exposure, which was already much higher than the ratio of D₂/H₂ measured by isotherms. If the D₂/H₂ by TDS increased to 9.5 for 120 min exposure time, what was the driving force for it? The authors didn't answer the question reasonably.

Answer: We would like to clarify that the equilibrium time for each point is not 8 minutes, especially in the low-pressure range. In the TDS measurements conducted with the H₂/D₂ mixture at 30 K, a pressure of 10 mbar (equivalent to 7.5 torr) was used. As mentioned in the response to the first question, it took 334 minutes, not 10 minutes, to reach 7.67 torr according to the isotherm data.

Furthermore, we would like to emphasize that the sorption isotherms were collected using single-component pure gases. Both H₂ and D₂ sorption isotherms were measured separately. However, in the TDS measurements, H₂/D₂ mixtures were used directly, and the competing effects of the two components must be taken into account.

Additionally, it is important to note that the measurement procedure of TDS indicates that this method operates under non-equilibrium conditions, as it involves the removal of non-adsorbed gas molecules from the system. As a result, it cannot be classified as an equilibrium-based method. For a more comprehensive explanation of this standard technique, please refer to the answer provided in the fourth question. Consequently, direct comparison between the two methods is not feasible.

Question 3: This reviewer has had concerns about the TDS test and thought TDS method used in this work might lead to an overestimated selectivity of D₂/H₂. As the adsorption/desorption kinetics were not very slow, the operation of gas molecules pump out could lead to a more serious problem for overestimated selectivity of D₂/H₂.

Answer: An overestimation of the selectivity by TDS can be confidently ruled out, as we have conducted numerous measurements on multiple samples, and all results have consistently shown high reproducibility. We have even repeated the measurements on fresh samples from the same batch to ensure the reliability of our findings. It is worth noting that many research papers from diverse groups have been published based on the technique of thermal desorption spectroscopy, illustrating its widespread usage and credibility. Here are a few examples:

- 1) Capture of heavy hydrogen isotopes in a metal-organic framework with active Cu(I) sites, *Nat. Commun.*, **2017**, 8, 14496.
- 2) In silico screening and experimental study of anion-pillared metal-organic frameworks for hydrogen isotope separation, *Sep. Purif. Technol.*, **2022**, 295, 121286.
- 3) Quantum sieving of H₂/D₂ in MOFs: a study on the correlation between the separation performance, pore size and temperature, *J. Mater. Chem. A*, **2020**, 8, 6319-6327.
- 4) Analysis of hydrogen isotopes with quadrupole mass spectrometry, *Anal. Methods*, **2017**, 9, 3067-3072.
- 5) Barely porous organic cages for hydrogen isotope separation, *Science*, **2019**, 366, 613-620.
- 6) Exploiting the specific isotope-selective adsorption of metal-organic framework for hydrogen isotope separation, *J. Am. Chem. Soc.*, **2021**, 143, 22, 8232–8236.
- 7) Highly effective hydrogen isotope separation through dihydrogen bond on Cu(I)-exchanged zeolites well above liquid nitrogen temperature, *Chem. Eng. J.*, **2020**, 391, 123485.

These published papers collectively demonstrate the use of various porous materials, including those with large and small pores, with or without uncoordinated metal sites, for isotope separation. Hence, TDS has proven to be a well-established and reliable technique for studying separation processes.

Question 4: The authors have not explained the question properly that why there are two distinct peaks of TDS for 30 K and 40 K, but only one for 50 K. Is it from D₂ gas impurities?

Answer: As we responded to the comment in the first run of reviewing, the two maxima depend on the exposure temperature. For a better and clearer understanding, we'd like to explain again the fundamental and procedure of thermal desorption spectroscopy in detail.

Thermal desorption spectroscopy (TDS), also known as temperature programmed desorption (TPD) is a standard surface science technique, provides information on the binding energies of atomic and molecular species adsorbed on a solid surface. The basic process of thermal desorption is that an adsorbate leaves the substrate and enters the

gas phase by transferring the thermal energy to the adsorbed species when the surface is heated, as shown in Figure R1. Thus, desorption takes place if a molecule has enough thermal energy to overcome the activation barrier for releasing the adsorbate from the substrate. Consequently, adsorbates that have a strong binding energy desorb at high temperatures, and adsorbates that have a weak binding energy desorb at low temperatures.

Figure R1. Illustration of adsorption and desorption process.

In a thermal desorption experiment for physisorbed hydrogen gases, Polanyi-Wigner equation follows first-order kinetics, implying that the desorption rate is proportional to instantaneous coverage, and the desorption peak will show a balance between coverage term and energy term. This is the usual desorption order for non-dissociative molecular adsorption, i.e., physisorption.

During a typical TDS measurement process, the sample is loaded onto a sample holder and subjected to vacuum activation to remove any potentially adsorbed molecules, such as impurities from sample loading or solvents from synthesis. This process is detected using a mass spectrometer. In the case of quantum sieving applications, a specific measurement procedure has been developed to optimize H₂/D₂ mixture separation, as outlined below:

1. An equimolar D₂/H₂ isotope mixture is loaded onto the sample at a fixed exposure temperature for a predetermined duration.
2. The free gas molecules are subsequently evacuated, and the sample is cooled down to 20 K to preserve the adsorbed state.
3. Finally, the sample is heated from 20 K to room temperature at a heating rate of 0.1 K/s, while the desorbing gas is continuously monitored using a mass spectrometer. A pressure increase in the sample chamber is observed when the gas desorbs, as illustrated in Figure R2.

Figure R2. TDS measurement procedure with isotope mixture.

As explained above, the residual gas molecules have been removed at the exposure temperature. In other words, during desorption, the gas molecules adsorbed with a lower activation energy than those adsorbed at the exposure temperature will leave the pores. Hence, in the case of 30 K, the gas molecules with both weak and stronger adsorbate-adsorbent interactions will stay on the surface, while at 50 K, only the gas molecules with a stronger interaction with the surface will stay. Therefore, at low temperatures of 30 and 40 K, two peaks can be observed, indicating two distinct desorption energy. Meanwhile, only one peak shown at 50 K represents that only the strongly bonded gas molecules stay, but the weakly bonded gas molecules have been removed already during the evacuation at the exposure temperature.

To summarize, the peaks only represent the interaction energy between adsorbent and adsorbate but are not related to the impurity. This phenomenon can also be observed in any gas species using TDS or TPD, e. g. CO_2 , O_2 , et al., not only for H_2 .

REVIEWERS' COMMENTS

Reviewer #3 (Remarks to the Author):

1. It is surprising to this reviewer that the authors used 99.8% D₂ directly to measure low capacity of D₂ adsorption without further purification.

2. In first round reviewing, this reviewer asked that "For each pressure step, the adsorption/desorption kinetic data should be collected and analyzed when discussing kinetic quantum sieving. If such data could not be obtained, at least the duration time to reach equilibrium for each step should be provided", but the authors didn't provide such data and only gave the total times for each measurement, including adsorption and desorption.

Temperature/K Equilibrium time/min

H₂ D₂

30 586.9 464.8

40 339.3 449.2

50 378.0 414.1

60 363.8 427.2

70 379.2 382.1

77 373.9 338.6

The 8 mins time for each step is an average value. It was calculated from the available data by this reviewer because the authors would not tell the duration time to reach equilibrium for each step. This time, the authors provided the measurement details of each data point for D₂ sorption at 30 K. It was true the initial several steps ($P < 1$ torr) required nearly 5 hours, but it does not mean that "a pressure of 10 mbar (equivalent to 7.5 torr) was used. As mentioned in the response to the first question, it took 334 minutes". In fact, when pressure from 2.444 torr to 7.677 torr, it only took 30 mins (from 303.7 to 334.2). Therefore, for a mixture of H₂ and D₂ exposure on USTC-700 for 120 mins at 10 mbar and 30 K, it could certainly reach an adsorption equilibrium. According to this work, the selectivity was 9.5 on USTC-700, which was ~ 8 times higher than the selectivity obtained from D₂ and H₂ single component gas adsorption measurement. Although the authors made a strong argument, but they did not give a scientific and reasonable reason to explain why it was so different from D₂, H₂ single gas measurement.

3. For the question "The authors have not explained the question properly that why there are two distinct peaks of TDS for 30 K and 40 K, but only one for 50 K. Is it from D₂ gas impurities?", the authors gave a long explanation but the core was "Hence, in the case of 30 K, the gas molecules with both weak and stronger adsorbate-adsorbent interactions will stay on the surface, while at 50 K, only the gas molecules with a stronger interaction with the surface will stay. Therefore, at low temperatures of 30 and 40 K, two peaks can be observed, indicating two distinct desorption energy. Meanwhile, only one peak shown at 50 K represents that only the strongly bonded gas molecules stay, but the weakly bonded gas molecules have been removed already during the evacuation at the exposure temperature". It is totally wrong because the sample was firstly cooled down to 20 K for all three exposure temperatures of 30K, 40K and 50K, and then the sample was heated from same 20 K to room temperature. The D₂ adsorption state should be same because it was re-created at 20 K.

RESPONSE TO REVIEWERS' COMMENTS

Reviewer #3 (Remarks to the Author):

Question 1: It is surprised to this reviewer that the authors used 99.8% D₂ directly to measure low capacity of D₂ adsorption without further purification.

Answer: The purity of 99.8% for D₂ is adequate for the related sorption and separation experiments in our work.

Question 2: In first round reviewing, this reviewer asked that “For each pressure steps, the adsorption/desorption kinetic data should be collected and analyzed when discussing kinetic quantum sieving. If such data could not be obtained, at least the duration time to reach equilibrium for each step should be provided”, but the authors didn't provide such data and only gave the total times for each measurement, including adsorption and desorption.

Temperature/K Equilibrium time/min

H₂ D₂

30 586.9 464.8

40 339.3 449.2

50 378.0 414.1

60 363.8 427.2

70 379.2 382.1

77 373.9 338.6

The 8 mins time for each step is an average value. It was calculated from the available data by this reviewer because the authors would not tell the duration time to reach equilibrium for each step. This time, the authors provided the measurement details of each data point for D₂ sorption at 30 K. It was true the initial several steps ($P < 1$ torr) required nearly 5 hours, but it does not mean that “a pressure of 10 mbar (equivalent to 7.5 torr) was used. As mentioned in the response to the first question, it took 334 minutes”. In fact, when pressure from 2.444 torr to 7.677 torr, it only took 30 mins (from 303.7 to 334.2). Therefore, for a mixture of H₂ and D₂ exposure on USTC-700 for 120 mins at 10 mbar and 30 K, it could certainly reach an adsorption equilibrium. According to this work, the selectivity was 9.5 on USTC-700, which was ~ 8 times higher than the selectivity obtained from D₂ and H₂ single component gas adsorption measurement. Although the authors made a strong argument, but they did not give a scientific and reasonable reason to explain why it was so different from D₂, H₂ single gas measurement.

Answer: We apologize that our explanation didn't fully clarify the terms related to hydrogen adsorption. For determining physisorption isotherms, the most common approach is volumetric method. In this case, a known amount of pure gas is dosed to a certain volume containing the adsorbent at a constant temperature. As adsorption takes place, the pressure in the volume drops until the equilibrium is established. The corresponding time is what we defined as equilibrium time. The pressure drop due to adsorption is then measured and another dose follows. This process is repeated until a full isotherm has been completed. On the other hand, thermal desorption spectroscopy (TDS), or temperature-programmed desorption (TPD), is also used as a technique in determine adsorption. It involves the desorption of the dosing gas into vacuum as a function of temperature, but not the difference of the dosing pressure. Therefore, distinct from volumetric method, TPD does not require to establish equilibrium. For small pore apertures, the diffusion of the gas molecules is limited. At low temperature, D_2 diffuse faster than H_2 molecules. Therefore, in a mixture measurement, the small channels lead to a single-file diffusion. In other words, once a D_2 molecule penetrated the pore, no passing between the isotopes in the channels is possible. Hence, the selectivity for mixture separation is much higher than simply compare the pure gas adsorption amount. We hope that our response will be helpful to understand those difference well.

Question 3: For the question "The authors have not explained the question properly that why there are two distinct peaks of TDS for 30 K and 40 K, but only one for 50 K. Is it from D_2 gas impurities?", the authors gave a long explanation but the core was "Hence, in the case of 30 K, the gas molecules with both weak and stronger adsorbate-adsorbent interactions will stay on the surface, while at 50 K, only the gas molecules with a stronger interaction with the surface will stay. Therefore, at low temperatures of 30 and 40 K, two peaks can be observed, indicating two distinct desorption energy. Meanwhile, only one peak shown at 50 K represents that only the strongly bonded gas molecules stay, but the weakly bonded gas molecules have been removed already during the evacuation at the exposure temperature". It is totally wrong because the sample was firstly cooled down to 20 K for all three exposure temperatures of 30K, 40K and 50K, and then the sample was heated from same 20 K to room temperature. The D_2 adsorption state should be same because it was re-created at 20 K.

Answer: We apologize that our explanation has been written in a complicated and misunderstood way. Here we try to formulate the answer in other words, which may be clearer. Figure 2c shows the TDS spectra for three different exposure temperatures, 30,

40, and 50 K. The sample is exposed to a 10 mbar H₂/D₂ mixture at the respective exposure temperature for 10 min. After 10 min gas exposure, the sample is kept at the exposure temperature, and the sample chamber is evacuated with a turbo-molecular pump, before finally cooling the sample down to 20 K in high vacuum with continuous pumping. For an exposure temperature of 50 K, the gas molecules adsorbed at stronger adsorption sites will remain adsorbed on the surface, whereas, the weakly bonded molecules are removed during evacuation at 50 K. Further cooling in high vacuum to 20 K does not lead to any redistribution of the adsorbed molecules on the surface, i.e. the molecules remain bonded on the stronger adsorption sites. Therefore, during the heating run in the TDS measurement, nearly no desorption is observed at temperatures below the exposure temperature. As a result, the low-temperature peak observed at T_{exp} = 30 K is decreased for T_{exp} = 40 K, and vanished for T_{exp} = 50 K. This technique of exposing the sample to gas at different exposure temperatures and evacuating at these temperatures has been applied for several porous materials to identify strong adsorption sites and their selectivity after H₂/D₂ mixture exposure, e.g. see Figure 5 in the paper of 10.1021/nn405420t, Figure 2 in the paper of 10.1038/ncomms14496, Figure 3 in the paper of 10.1021/acs.inorgchem.2c00028. Gas impurities as a possible reason can be excluded since impurity gases would have been detected by the mass spectrometer.